# SEQUENTIAL LATENT KNOWLEDGE SELECTION FOR KNOWLEDGE-GROUNDED DIALOGUE

**Byeongchang Kim**    **Jaewoo Ahn**    **Gunhee Kim**
Department of Computer Science and Engineering
Seoul National University, Seoul, Korea
{byeongchang.kim,jaewoo.ahn}@vision.snu.ac.kr gunhee@snu.ac.kr
http://vision.snu.ac.kr/projects/skt

## ABSTRACT

Knowledge-grounded dialogue is a task of generating an informative response based on both discourse context and external knowledge. As we focus on better modeling the *knowledge selection* in the multi-turn knowledge-grounded dialogue, we propose a sequential latent variable model as the first approach to this matter. The model named *sequential knowledge transformer* (SKT) can keep track of the prior and posterior distribution over knowledge; as a result, it can not only reduce the ambiguity caused from the diversity in knowledge selection of conversation but also better leverage the response information for proper choice of knowledge. Our experimental results show that the proposed model improves the knowledge selection accuracy and subsequently the performance of utterance generation. We achieve the new state-of-the-art performance on Wizard of Wikipedia (Dinan et al., 2019) as one of the most large-scale and challenging benchmarks. We further validate the effectiveness of our model over existing conversation methods in another knowledge-based dialogue Holl-E dataset (Moghe et al., 2018).

## 1 INTRODUCTION

Knowledge-grounded dialogue is a task of generating an informative response based on both discourse context and selected external knowledge (Ghazvininejad et al., 2018). For example, it is more descriptive and engaging to respond *"I've always been more of a fan of the American football team from Pittsburgh, the Steelers!"* than *"Nice, I like football too."* (Dinan & Weston, 2019). As it has been one of the key milestone tasks in conversational research (Zhang et al., 2018), a majority of previous works have studied how to effectively combine given knowledge and dialogue context to generate an utterance (Zhang et al., 2018; Li et al., 2019b; Parthasarathi & Pineau, 2018; Madotto et al., 2018; Gopalakrishnan et al., 2019). Recently, Dinan et al. (2019) proposed to tackle the knowledge-grounded dialogue by decomposing it into two sub-problems: first selecting knowledge from a large pool of candidates and generating a response based on the selected knowledge and context.

In this work, we investigate the issue of *knowledge selection* in the multi-turn knowledge-grounded dialogue, since practically the selection of pertinent topics is critical to better engage humans in conversation, and technically the utterance generation becomes easier with a more powerful and consistent knowledge selector in the system. Especially, we focus on developing a *sequential latent variable model* for knowledge selection, which has not been discussed in previous research. We believe it brings several advantages for more engaging and accurate knowledge-based chit-chat. First, it can correctly deal with the *diversity* in knowledge selection of conversation. Since one can choose any knowledge to carry on the conversation, there can be one-to-many relations between dialogue context and knowledge selection. Such multimodality by nature makes the training of a dialogue system much more difficult in a data-driven way. However, if we can sequentially model the history of knowledge selection in previous turns, we can reduce the scope of probable knowledge candidates at current turn. Second, the sequential latent model can better leverage the response information, which makes knowledge selection even more accurate. It is naturally easy to select the knowledge in the pool once the response is known, because the response is generated based on the

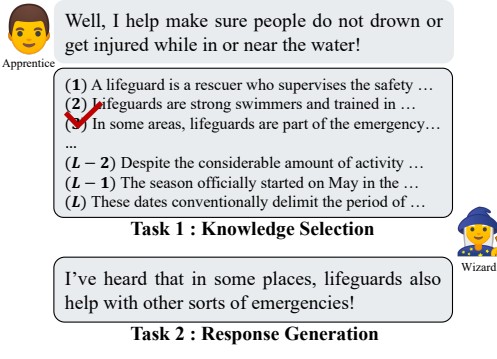

Figure 1: An example of wizard's tasks in knowledge-grounded conversation of Wizard of Wikipedia (Dinan et al., 2019).

Table 1: Accuracy of knowledge selection with and without knowing the response. We test with GRU (Cho et al., 2014), Transformer (Vaswani et al., 2017) and BERT (Devlin et al., 2019) as the sentence encoder. For human evaluation, we randomly sample 20 dialogues and ask human annotators to select the most likely knowledge sentence from the pool.

| Methods | w/o response | w/ response |
|---|---|---|
| GRU | 20.0 | 66.0 |
| Transformer | 22.5 | 70.4 |
| BERT | 23.4 | 78.2 |
| Transformer + GT history | 25.4 | 70.4 |
| BERT + GT history | 27.3 | 79.2 |
| Random | 2.7 | 2.7 |
| Human | 17.1 | 83.7 |

selected knowledge. Our sequential model can keep track of prior and posterior distribution over knowledge, which are sequentially updated considering the responses in previous turns, and thus we can better predict the knowledge by sampling from the posterior. Third, the latent model works even when the knowledge selection labels for previous dialogue are not available, which is common in practice. For example, if multiple people have discussion about given documents, knowledge selection of previous turns is done by others. The latent model can infer which knowledge others are likely to select and use.

Finally, the contributions of this work are as follows.

1. We propose a novel model named *sequential knowledge transformer* (SKT). To the best of our knowledge, our model is the first attempt to leverage a sequential latent variable model for knowledge selection, which subsequently improves knowledge-grounded chit-chat.

2. Our experimental results show that the proposed model improves not only the knowledge selection accuracy but also the performance of utterance generation. As a result, we achieve the new state-of-the-art performance on Wizard of Wikipedia (Dinan et al., 2019) and a knowledge-annotated version of Holl-E (Moghe et al., 2018) dataset.

## 2 PROBLEM STATEMENT AND MOTIVATION

As a main testbed of our research, we choose the Wizard of Wikipedia (WoW) benchmark (Dinan et al., 2019), since it is one of the most large-scale and challenging datasets for open-domain multi-turn knowledge-based dialogue. Moreover, the dataset can evaluate the algorithm's ability for solving the two subproblems of knowledge selection and response generation. That is, it provides ground-truth labels of knowledge selection and clear grounding between the pairs of selected knowledge and response. In our experiments, we also evaluate on Holl-E (Moghe et al., 2018) as another dataset for knowledge-grounded dialogue, after collecting clearer labels of knowledge sentences.

**The Flow of Conversation**. The WoW (Dinan et al., 2019) deals with a chit-chat dialogue task where two speakers discuss in depth about a given topic. One speaker (coined as *Wizard*) is to be both engaging and knowledgeable on the topic with access to an information retrieval (IR) system over Wikipedia to supplement its knowledge. The other speaker (*Apprentice*) is curious and eager to learn about the topic. With an example in Figure 1, the conversation flow takes place as follows.

1. One topic is chosen among 1,431 topics and shared between the two speakers.

2. Given an apprentice's utterance and a wizard's previous utterance, the IR system retrieves relevant knowledge, which includes the first paragraph of top 7 articles each for wizard and apprentice and the first 10 sentences of the original Wikipedia page of the topic (*e.g.* the *lifeguard* wikipage). The knowledge pool contains 67.57 sentences on average. Then.

the wizard must choose a single relevant sentence from them (*knowledge selection*) and construct an utterance (*response generation*).

3. The conversation repeats until a minimum number of turns (5 each) reaches.

**The Motivation of Sequential Latent Models**. The goal of the task is to model the wizard that solves the two subproblems of knowledge selection and response generation (Dinan et al., 2019). In the knowledge selection step, a single relevant knowledge sentence is chosen from a pool of candidates, and in the response generation step, a final utterance is generated with the chosen knowledge and dialogue context. This pipeline is originally proposed to tackle open-domain TextQA (Chen et al., 2017); for example, Min et al. (2018) show its effectiveness for single-document TextQA, to which the key is to locate the sentences that contain the information about the answer to a question.

For knowledge-grounded dialogue, however, there can be one-to-many relations between the dialogue context and the knowledge to be selected unlike TextQA. Except a direct question about context, one can choose any diverse knowledge to carry on the conversation. Therefore, the knowledge selection in dialogue is diverse (*i.e.* multimodal) by nature, which should be correctly considered in the model. It is our main motivation to propose a sequential latent variable model for knowledge selection, which has not been studied yet. The latent variable not only models such diversity of knowledge but also sequentially track the topic flow of knowledge in the multi-turn dialogue.

Another practical advantage of the sequential latent model lies in that it is easy to find which knowledge is chosen once the response is known, since the response is written based on the selected knowledge. Table 1 clearly validates this relation between knowledge and response. In the WoW dataset, knowing a response boosts the accuracy of knowledge sentence selection for both human and different models. These results hint that knowledge selection may need to be jointly modeled with response generation in a sequence of multi-turn chit-chats, which can be done by the sequential latent models.

## 3 APPROACH

We propose a novel model for knowledge-grounded conversation named *sequential knowledge transformer* (SKT), whose graphical model is illustrated in Figure 2. It is a sequential latent model that sequentially conditions on previously selected knowledge to generate a response.

We will use $1 \leq t \leq T$ to iterate over dialogue turns, $1 \leq m \leq M$ and $1 \leq n \leq N$ to respectively iterate over words in the utterance of apprentice and wizard, and $1 \leq l \leq L$ to denote knowledge sentences in the pool. Thus, $T$ is the dialogue length, $M$ and $N$ are the length of each utterance of apprentice and wizard, and $L$ is the size of the knowledge pool.

The input to our model at turn $t$ is previous turns of conversation, which consists of utterances from apprentice $\mathbf{x}^1, ..., \mathbf{x}^t$, utterances from wizard $\mathbf{y}^1, ..., \mathbf{y}^{t-1}$ and the knowledge pool $\mathbf{k}^1, ..., \mathbf{k}^t$, where $\mathbf{k}^t = \{\mathbf{k}^{t,l}\} = \mathbf{k}^{t,1}, ..., \mathbf{k}^{t,L}$. The output of the model is selected knowledge $\mathbf{k}_s^t$ and the wizard's response $\mathbf{y}^t$. Below, we discuss sentence embedding, knowledge selection and utterance decoding in our approach. Note that our technical novelty lies in the knowledge selection model, while exploiting existing techniques for text encoding and utterance decoding.

**Sentence Encoding**. We represent an apprentice utterance $\mathbf{x}^t$ to an embedding $\mathbf{h}_x^t$ using BERT (Devlin et al., 2019) and average pooling over time steps (Cer et al., 2018):

$$\mathbf{H}_x^t = \text{BERT}_{base}([x_1^t; ...; x_M^t]) \in \mathbb{R}^{M \times 768}, \mathbf{h}_x^t = \text{avgpool}(\mathbf{H}_x^t) \in \mathbb{R}^{768}. \tag{1}$$

Likewise, the utterance of Wizard $\mathbf{y}^{t-1}$ is embedded as $\mathbf{h}_y^{t-1}$ and knowledge sentences are as $\{\mathbf{h}_k^{t,l}\} = \mathbf{h}_k^{t,1}, ..., \mathbf{h}_k^{t,L}$. Each apprentice-wizard utterance pair $\mathbf{h}_{xy}^t = [\mathbf{h}_x^t; \mathbf{h}_y^t]$ at dialog turn $t$ is jointly represented through a GRU (Cho et al., 2014) layer: $\mathbf{d}_{xy}^t = \text{GRU}_{dialog}(\mathbf{d}_{xy}^{t-1}, \mathbf{h}_{xy}^t) \in \mathbb{R}^{768}$.

**Sequential Knowledge Selection**. Compared to previous works, we make two significant modifications. First, we regard the knowledge selection as a sequential decision process instead of a single-step decision process. Second, due to the diversity of knowledge selection in dialogue, we model it as latent variables. As a result, we can carry out the joint inference of multi-turns of knowledge selection and response generation rather than separate inference turn by turn.

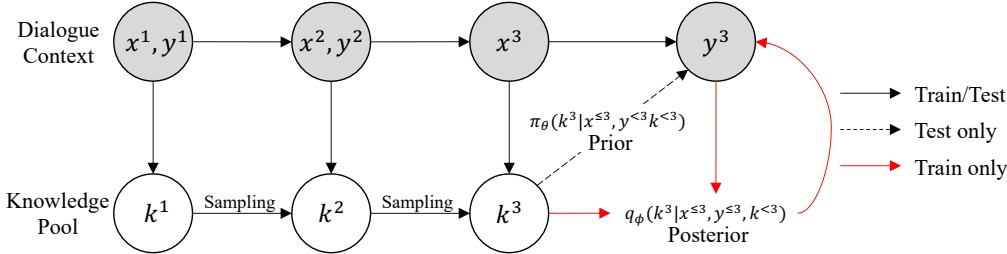

Figure 2: A graphical representation of the proposed *sequential knowledge transformer* (SKT) model. At the third turn, the goal is to generate wizard's response ($\mathbf{y}^3$) given dialogue context ($\mathbf{x}^{\leq 3}, \mathbf{y}^{<3}$). Our model sequentially infer which knowledge is likely to be used ($\mathbf{k}^{\leq 3}$), from which the utterance $\mathbf{y}^3$ is generated.

There have been much research on sequential latent variable models (Chung et al., 2015; Fraccaro et al., 2016; Goyal et al., 2017; Aneja et al., 2019; Shankar & Sarawagi, 2019). For example, Shankar & Sarawagi (2019) propose a posterior attention model that represents the attention of seq2seq models as sequential latent variables. Inspired by them, we factorize the response generation with latent knowledge selection and derive the variational lower bound as follows:

$$\log p(\mathbf{y}|\mathbf{x}) = \log \prod_t \sum_{\mathbf{k}^t} p_\theta(\mathbf{y}^t|\mathbf{x}^{\leq t}, \mathbf{y}^{<t}, \mathbf{k}^{\leq t}) \pi_\theta(\mathbf{k}^t|\mathbf{x}^{\leq t}, \mathbf{y}^{<t}, \mathbf{k}^{<t}) \tag{2}$$

$$\geq \sum_t \mathbb{E}_{q_\phi(\mathbf{k}^{t-1})} \Big[ \mathbb{E}_{q_\phi(\mathbf{k}^t)} [\log p_\theta(\mathbf{y}^t|\mathbf{x}^{\leq t}, \mathbf{y}^{<t}, \mathbf{k}^t)] - D_{KL}(q_\phi(\mathbf{k}^t) \| \pi_\theta(\mathbf{k}^t)) \Big], \tag{3}$$

where $q_\phi(\mathbf{k}^t)$ is shorthand for $q_\phi(\mathbf{k}^t|\mathbf{x}^{\leq t}, \mathbf{y}^{\leq t}, \mathbf{k}^{<t})$ and $\pi_\theta(\mathbf{k}^t)$ for $\pi_\theta(\mathbf{k}^t|\mathbf{x}^{\leq t}, \mathbf{y}^{<t}, \mathbf{k}^{<t})$ for brevity. Note that $p_\theta(\mathbf{y}^t|\cdot)$ is a decoder network, $\pi_\theta(\mathbf{k}^t)$ is a categorical conditional distribution of knowledge given dialogue context and previously selected knowledge, and $q_\phi(\mathbf{k}^t)$ is an inference network to approximate posterior distribution $p_\theta(\mathbf{k}^t|\mathbf{x}^{\leq t}, \mathbf{y}^{\leq t}, \mathbf{k}^{<t})$.

The conditional probability of generating wizard's response $\mathbf{y}^t$ given dialogue context $\mathbf{x}^{\leq t}$ and $\mathbf{y}^{<t}$, can be re-written from Eq. (2) as follows:

$$p(\mathbf{y}^t|\mathbf{x}^{\leq t}, \mathbf{y}^{<t}) \approx \prod_{i=1}^{t-1} \sum_{\mathbf{k}^i} q_\phi(\mathbf{k}^i) \Big( \sum_{\mathbf{k}^t} p_\theta(\mathbf{y}^t|\mathbf{x}^{\leq t}, \mathbf{y}^{<t}, \mathbf{k}^t) \pi_\theta(\mathbf{k}^t) \Big). \tag{4}$$

The detailed derivation can be found in Appendix. Eq.(4) means that we first infer from the knowledge posterior which knowledge would be used up to previous turn $t-1$, estimate the knowledge for current turn $t$ from prior knowledge distribution and generate an utterance from the inferred knowledge. Figure 2 shows an example of this generation process at $t = 3$. We parameterize the decoder network $p_\theta$, the prior distribution of knowledge $\pi_\theta$, and the approximate posterior $q_\phi$ with deep neural networks as will be discussed.

From the posterior distribution $q_\phi(\mathbf{k}^{t-1})$ we draw a sample $\mathbf{k}_s^{t-1}$, and then update $\pi_\theta$ and $q_\phi$ with the sentence embedding of sampled knowledge ($\mathbf{h}_k^{t-1,s}$) and the embeddings of previous and current utterances ($\mathbf{d}_{xy}^{t-1}, \mathbf{d}_{xy}^t, \mathbf{h}_x^t$). We use an attention mechanism over current knowledge pool $\{\mathbf{h}_k^{t,l}\}$ to compute knowledge distribution given the dialogue context. This process is modeled as

$$\pi_\theta(\mathbf{k}^t|\mathbf{x}^{\leq t}, \mathbf{y}^{<t}, \mathbf{k}_s^{\leq t-1}) = \mathrm{softmax}(\mathbf{q}_{prior}^t[\mathbf{h}_k^{t,1}, ..., \mathbf{h}_k^{t,L}]^\top) \in \mathbb{R}^L \tag{5}$$

$$q_\phi(\mathbf{k}^t|\mathbf{x}^{\leq t}, \mathbf{y}^{\leq t}, \mathbf{k}_s^{\leq t-1}) = \mathrm{softmax}(\mathbf{q}_{post}^t[\mathbf{h}_k^{t,1}, ..., \mathbf{h}_k^{t,L}]^\top) \in \mathbb{R}^L, \tag{6}$$

where

$$\mathbf{q}_{prior}^t = \mathbf{W}_{prior}([\mathbf{d}_{xy}^{t-1}; \mathbf{h}_x^t; \mathrm{GRU}_{hist}(\mathbf{d}_k^{t-2}, \mathbf{h}_k^{t-1,s})]), \tag{7}$$

$$\mathbf{q}_{post}^t = \mathbf{W}_{post}([\mathbf{d}_{xy}^t; \mathrm{GRU}_{hist}(\mathbf{d}_k^{t-2}, \mathbf{h}_k^{t-1,s})]), \tag{8}$$

$\mathbf{d}_k^t$ is the hidden state of $\mathrm{GRU}_{hist}$ and we initialize $\mathbf{d}_{xy}^0 = \mathbf{d}_k^0 = \mathbf{0} \in \mathbb{R}^{768}$, and $\mathbf{W}_{prior}, \mathbf{W}_{post} \in \mathbb{R}^{768 \times (768*2)}$ are the parameters. We here use the GRU (Li et al., 2017; Aneja et al., 2019) to sequentially condition previously selected knowledge to $\pi_\theta$ and $q_\phi$.

Finally, we sample knowledge $\mathbf{k}_s^t$ over attention distribution in Eq. (6) and pass it to the decoder. At test time, we select the knowledge with the highest probability over distribution in Eq. (5).

**Decoding with Copy Mechanism**. We generate the wizard's response at turn $t$, given current context $\mathbf{x}^t$ and selected knowledge sentence $\mathbf{k}_s^t$. We feed their concatenated embedding $\mathbf{H}_{xk_s}^t = [\mathbf{H}_x^t; \mathbf{H}_{k_s}^t]$ to the decoder $p_\theta$. To maximize the effect of selected knowledge for response generation, we choose the Copy mechanism (Xia et al., 2017; Li et al., 2019b) with Transformer decoder (Vaswani et al., 2017). We obtain the output word probability (Zhao et al., 2019a):

$$\mathbf{h}_n^t = \text{Decoder}(\mathbf{H}_{xk_s}^t, \mathbf{y}_{<n}^t), \quad \mathbf{q}_n^t, \mathbf{K}^t, \mathbf{V}^t = \mathbf{h}_n^t \mathbf{W}_q^\top, \mathbf{H}_{xk_s}^t \mathbf{W}_k^\top, \mathbf{H}_{xk_s}^t \mathbf{W}_v^\top, \tag{9}$$

$$p_{t,n}^{gen}(w) = \text{softmax}(\mathbf{W}_{out}\mathbf{h}_n^t), \quad p_{t,n}^{copy}(w) = \text{softmax}(\mathbf{q}_n^t \mathbf{K}^t), \tag{10}$$

$$p_{t,n}(w) = (1 - \alpha_{t,n}^{copy}) * p_{t,n}^{gen}(w) + \alpha_{t,n}^{copy} * p_{t,n}^{copy}(w), \tag{11}$$

where $\alpha_{t,n}^{copy} = \sigma(\mathbf{W}_{copy}^\top \sum p_{t,n}^{copy}(w) \cdot \mathbf{V}^t)$ and $\sigma$ is a sigmoid. Finally, we select the word with the highest probability $y_{n+1}^t = \arg\max_{w \in \mathcal{V}} p_{t,n}(w)$ where $\mathcal{V}$ is the dictionary. Unless the word $y_{n+1}^t$ is an EOS token, we repeat generating the next word by feeding $y_{n+1}^t$ to the decoder.

## 3.1 Training

Obviously, there is a large gap in knowledge selection accuracy between training with or without true labels (*e.g.* 23.2 of E2E Transformer MemNet with labels vs 4.8 of PostKS without labels in Table 2). As one way to take advantage of true labels for training of latent models, prior research has employed auxiliary losses over latent variables (Wen et al., 2017; Zhao et al., 2017). Similarly, we use the knowledge loss from Dinan et al. (2019) (*i.e.* the cross-entropy loss between predicted and true knowledge sentences) as an auxiliary loss for the latent variable. Thus, the training objective is a combination of the variational lower-bound from Eq. (3) and the auxiliary knowledge loss as

$$\mathcal{L} = -\frac{1}{T} \sum_{t=1}^{T} \mathbb{E}_{q_\phi(\mathbf{k}^{t-1})} \Big[ \mathbb{E}_{q_\phi(\mathbf{k}^t)}[\log p_\theta(\mathbf{y}^t | \mathbf{x}^{\leq t}, \mathbf{y}^{<t}, \mathbf{k}_s^t)]$$

$$- D_{KL}(q_\phi(\mathbf{k}^t) \| \pi_\theta(\mathbf{k}^t)) + \lambda \underbrace{\log q_\phi(\mathbf{k}_a^t)}_{\text{Knowledge loss}} \Big], \tag{12}$$

where $\mathbf{k}_s^t$ is a sampled knowledge from $q_\phi(\mathbf{k}^t | \mathbf{x}^{\leq t}, \mathbf{y}^{\leq t}, \mathbf{k}^{<t})$, $\mathbf{k}_a^t$ is a true knowledge, and $\lambda$ is a hyperparameter. Note that knowledge is sequentially sampled from attention distribution as in Eq. (6). We train our model by mini-batch gradient descent. We approximate the expectation by drawing one sample from the posterior with Gumbel-Softmax function (Jang et al., 2017; Maddison et al., 2017b). Further details of optimization can be found in Appendix.

## 4 Experiments

We evaluate our model mainly on the Wizard of Wikipedia (Dinan et al., 2019) and additionally Holl-E (Moghe et al., 2018) as another knowledge-grounded chit-chat dataset. We quantitatively and qualitatively compare our approach with other state-of-the-art models.

## 4.1 Datasets

**Wizard of Wikipedia**. It contains 18,430 dialogues for training, 1,948 dialogues for validation and 1,933 dialogues for test. The test set is split into two subsets, *Test Seen* and *Test Unseen*. Test Seen contains 965 dialogues on the topics overlapped with the training set, while Test Unseen contains 968 dialogues on the topics never seen before in training and validation set.

**Holl-E**. It contains 7,228 dialogues for training, 930 dialogues for validation and 913 dialogues for test. A single document is given per dialogue; the documents include about 58 and 63 sentences on average for training/validation and test set, respectively. The dataset provides *spans* in the document as additional information to provide which parts of the document is used to generate a response. However, the span labels are rather inconsistent; for example, they are often shorter than a single sentence or contain multiple consecutive sentences. Thus, we collect a new set of ground-truth (GT)

Table 2: Quantitative results on the Wizard of Wikipedia dataset (Dinan et al., 2019). The method with [*] does not use the knowledge loss. The scores of E2E Transformer MemNet[†] and Transformer (no knowledge)[†] are from the original paper. The variant (BERT vocab)[‡] is re-runned using the authors' code, since the vocabulary is different from original paper due to the use of BERT.

| | Test Seen | | | | Test Unseen | | | |
|---|---|---|---|---|---|---|---|---|
| Method | PPL | R-1 | R-2 | Acc | PPL | R-1 | R-2 | Acc |
| Random knowledge selection | - | 8.4 | 1.4 | 2.7 | - | 8.0 | 1.2 | 2.3 |
| Repeat last utterance | - | 14.5 | 3.1 | - | - | 14.1 | 2.9 | - |
| Transformer (no knowledge)[†] (Dinan et al., 2019) | **41.8** | 17.8 | - | - | 87.0 | 14.0 | - | - |
| E2E Transformer MemNet[†] (Dinan et al., 2019) | 63.5 | 16.9 | - | 22.5 | 97.3 | 14.4 | - | 12.2 |
| E2E Transformer MemNet (BERT vocab)[‡] | 53.2 | 17.7 | 4.8 | 23.2 | 137.8 | 13.6 | 1.9 | 10.5 |
| PostKS* (Lian et al., 2019) | 79.1 | 13.0 | 1.0 | 4.8 | 193.8 | 13.1 | 1.0 | 4.2 |
| E2E BERT | 53.5 | 16.8 | 4.5 | 23.7 | 105.7 | 13.5 | 2.2 | 13.6 |
| PostKS + Knowledge Loss | 54.5 | 18.1 | 5.3 | 23.4 | 144.8 | 13.5 | 2.0 | 9.4 |
| E2E BERT + PostKS | 54.6 | 17.8 | 5.3 | 25.5 | 113.2 | 13.4 | 2.3 | 14.1 |
| E2E BERT + PostKS + Copy | 52.2 | 19.0 | 6.5 | 25.5 | 83.4 | 15.6 | 3.9 | 14.4 |
| Ours | 52.0 | **19.3** | **6.8** | **26.8** | **81.4** | **16.1** | **4.2** | **18.3** |

Table 3: Quantitative results on the Holl-E dataset (Moghe et al., 2018) with single reference and multiple references test set.

| | Single Reference | | | | Multiple References | | | |
|---|---|---|---|---|---|---|---|---|
| Method | PPL | R-1 | R-2 | Acc | PPL | R-1 | R-2 | Acc |
| Random knowledge selection | - | 7.4 | 1.8 | 1.9 | - | 10.3 | 3.6 | 3.5 |
| Repeat last utterance | - | 11.4 | 1.5 | - | - | 13.6 | 2.0 | - |
| E2E Transformer MemNet (Dinan et al., 2019) | 140.6 | 20.1 | 10.3 | 22.7 | 83.6 | 24.3 | 12.8 | 32.3 |
| PostKS* (Lian et al., 2019) | 196.6 | 15.2 | 6.0 | 1.5 | 114.1 | 19.2 | 7.9 | 3.2 |
| E2E BERT | 112.6 | 25.9 | 18.3 | 28.2 | 66.9 | 31.1 | 22.7 | 37.5 |
| PostKS + Knowledge Loss | 135.1 | 19.9 | 10.7 | 22.5 | 81.9 | 23.8 | 12.9 | 32.2 |
| E2E BERT + PostKS | 119.9 | 27.8 | 20.1 | 27.6 | 66.7 | 33.7 | 25.8 | 37.3 |
| E2E BERT + PostKS + Copy | **47.4** | 29.2 | 22.3 | 27.8 | **27.9** | 35.9 | 29.0 | 37.8 |
| Ours | 48.9 | **29.8** | **23.1** | **29.2** | 28.5 | **36.5** | **29.7** | **39.2** |

knowledge per document so that it is similar to that of WoW where all of the GT knowledge are in the form of sentences. Basically, we select the sentence that includes the span as the GT knowledge sentence. If the span is given over multiple sentences, we select the minimum number of consecutive sentences containing the span as GT. If no span is given, we use the *no passages used* tag as GT, which amounts to 5% of all GT labels. It indicates that the gold utterance is generated with no knowledge grounding and the model should predict the label of *no passages used* for this sample to be correct. We make our new set of GT annotations available in the project page.

## 4.2 EXPERIMENTAL SETTING

**Evaluation Metrics**. We follow the evaluation protocol of WoW (Dinan et al., 2019). We measure unigram F1 (R-1), bigram F1 (R-2) and perplexity (PPL) for response generation, and the accuracy for knowledge selection. For *n*-gram metrics, we remove all the punctuations and (a, an, the) before computing the score. We remind that lower perplexity and higher *n*-gram (R-1, R-2) scores indicate better performance.

The test set for Holl-E is split into two subsets, *single reference* and *multiple references*. The dataset basically provides a single response per context (denoted as single reference). However, for some conversations, more responses (*e.g.* 2–13) are collected from multiple annotators per context (multiple references). For evaluation of multiple references, we take the best score over multiple GTs by following Moghe et al. (2018). For knowledge accuracy, we regard the model's prediction is correct if it matches at least one of the correct answers.

**Baselines**. We closely compare with two state-of-the-art knowledge-grounded dialogue models. The first one is E2E Transformer MemNet (Dinan et al., 2019), which uses a Transformer memory network for knowledge selection and a Transformer decoder for utterance prediction. The second

one is PostKS (Lian et al., 2019), which uses the posterior knowledge distribution as a pseudo-label for knowledge selection. For fair comparison, we replace all GRU layers in PostKS with Transformers. We also compare with four variants of these models as an ablation study: (i) E2E BERT, where we replace the Transformer memory network with pre-trained BERT, (ii) PostKS + Knowledge loss, where we additionally use the knowledge loss, (iii) E2E BERT + PostKS, which combines all the components of baselines, and (iv) E2E BERT + PostKS + Copy, where we additionally use the copy mechanism with the Transformer decoder.

We use official BERT tokenizer to tokenize the words and use pre-defined BERT vocabulary ($\mathcal{V} = 30522$) to convert token to index[1]. All the baselines use the exactly same inputs with our model except PostKS, which does not make use of knowledge labels as proposed in the original paper.

### 4.3 QUANTITATIVE RESULTS

Table 2 compares the performance of different methods on the Wizard of Wikipedia dataset. Our model outperforms the state-of-the-art knowledge-grounded dialogue models in all metrics for knowledge selection (accuracy) and utterance generation (unigram F1, bigram F1). The PostKS that is trained with no knowledge label shows low accuracy on knowledge selection, which is slightly better than random guess. However, it attains better performance than E2E Transformer MemNet with the knowledge loss in the WoW Test Seen, which shows that leveraging prior and posterior knowledge distribution is effective for knowledge-grounded dialogue, although using sequential latent variable improves further. BERT improves knowledge selection accuracy, but not much as in TextQA because of diversity in knowledge selection of conversation. The E2E BERT + PostKS + Copy performs the best among baselines, but not as good as ours, which validates that sequential latent modeling is critical for improving the accuracy of knowledge selection and subsequently utterance generation. Additionally, the performance gaps between ours and baselines are larger in Test Unseen. It can be understood that the sequential latent variable can generalize better. Adding the copy mechanism to the baseline substantially improves the accuracy of utterance generation, but barely improves the knowledge selection, which also justifies the effectiveness of the sequential latent variable. Transformer (no knowledge) shows the lowest perplexity in the WoW Test Seen, and it is mainly due to that it may generate only general and simple utterances since no knowledge is grounded. This behavior can be advantageous for the perplexity, while the other knowledge-based models take a risk of predicting wrong knowledge, which is unfavorable for perplexity.

Table 3 compares the performance of our model on Holl-E dataset. Similarly, our model outperforms all the baselines in all metrics. One notable trend is that BERT considerably reduces the perplexity in all models, which may be due to that the dataset size of Holl-E is much smaller than WoW and BERT prevents overfitting (Hao et al., 2019).

### 4.4 QUALITATIVE RESULTS

**Single-Turn Human Evaluation**. We perform a user study to complement the limitation of automatic language metrics. We evaluate several aspects of utterance generation using the similar setting in Guu et al. (2018). We randomly sample 100 test examples, and each sample is evaluated by three unique human annotators on Amazon Mechanical Turk (AMT). At test, we show dialogue context and generated utterance by our method or baselines. We ask turkers to rate the quality of each utterance in two aspects, which are referred to Li et al. (2019a): (i) *Engagingness*: how much do you like the response? and (ii) *Knowledgeability*: how much is the response informative? Each item is scored from 1 to 4 to avoid *catch-all* category in the answer (Dalal et al., 2014), where 1 means *not at all*, 2 is *a little*, 3 is *somewhat*, and 4 is *a lot*. To mitigate annotator bias and inter-annotator variability, we adjust human scoring with Bayesian calibration (Kulikov et al., 2019). Note that human evaluation on knowledge selection is not possible, since any knowledge could be fine for a given context, which is key motivation for our sequential latent model – *diversity* of knowledge selection.

Table 4 summarizes the results of the single-turn human evaluation, which validates that annotators prefer our results to those of baselines. Again, the performance gaps between ours and baselines are larger in Test Unseen, thank to better generality of our sequential latent model.

---

[1]https://github.com/tensorflow/models/tree/master/official/nlp/bert.

Table 4: Single-turn human evaluation results on the Wizard of Wikipedia. We report the mean ratings and their standard errors of different methods for engagingness and knowledgeability scores. TMN stands for E2E Transformer MemNet (Dinan et al., 2019).

| | Test Seen | | | | Test Unseen | | | |
| | Raw | | Calibrated | | Raw | | Calibrated | |
| Method | Engage | Knowledge | Engage | Knowledge | Engage | Knowledge | Engage | Knowledge |
|---|---|---|---|---|---|---|---|---|
| PostKS | 1.65 (0.05) | 1.72 (0.06) | 1.51 (0.02) | 1.72 (0.01) | 1.66 (0.06) | 1.74 (0.06) | 1.38 (0.02) | 1.60 (0.02) |
| TMN | 2.57 (0.05) | 2.47 (0.06) | 2.41 (0.02) | 2.49 (0.01) | 2.39 (0.06) | 2.21 (0.06) | 2.12 (0.02) | 2.05 (0.02) |
| Ours | 2.59 (0.05) | 2.53 (0.06) | 2.45 (0.02) | 2.55 (0.01) | 2.52 (0.06) | 2.35 (0.06) | 2.26 (0.02) | 2.21 (0.02) |
| Human | 3.14 (0.05) | 3.09 (0.05) | 3.00 (0.02) | 3.12 (0.01) | 3.11 (0.05) | 2.99 (0.05) | 2.83 (0.01) | 2.85 (0.02) |

Table 5: Multi-turn human evaluation results on the Wizard of Wikipedia. We report the averages and standard deviations (in parentheses).

| Method | Test Seen | Test Unseen |
|---|---|---|
| E2E Transformer MemNet (Dinan et al., 2019) | 2.36 (1.38) | 2.10 (0.96) |
| Ours | 2.39 (0.99) | 2.38 (1.01) |
| Human (Dinan et al., 2019) | 4.13 (1.08) | 4.34 (0.98) |

**Multi-turn Human Evaluation**. We add another human evaluation results in a multi-turn setting using the evaluation toolkit from Wizard of Wikipedia (Dinan et al., 2019). Humans are paired with one of the models and chat about a specific topic (given a choice of 2–3 topics) for 3–5 dialogue turns. After conversation, they score their dialogue partners on a scale of 1–5, with the rating indicating how much they *liked* the conversation. We collect the votes for 110 randomly sampled conversations from 11 different turkers.

Table 5 compares the results of different methods for the multi-turn evaluation. Human annotators prefer our results to those of baselines with a larger gap in Test Unseen.

**Dialogue Examples**. Figure 3 shows selected examples of utterance prediction. In each set, we show dialogue context, human response, and utterances generated by our method and baselines. Thanks to the use of latent variables, our model can better capture the changes in dialogue topics and thus generate more appropriate responses.

## 5 RELATED WORK

Knowledge-based conversations have been studied much including collecting new datasets (Qin et al., 2019; Zhang et al., 2018; Ghazvininejad et al., 2018; Zhou et al., 2018; Dinan et al., 2019; Moghe et al., 2018) or developing new models (Lian et al., 2019; Li et al., 2019b; Yavuz et al., 2019; Zhao et al., 2019b; Dinan et al., 2019; Liu et al., 2019). Most works on the models have less investigated the knowledge selection issue but instead focused on how to effectively combine given knowledge and dialogue context to improve response informativeness. For example, Ghazvininejad et al. (2018) aid a Seq2Seq model with an external knowledge memory network, and Li et al. (2019b) propose an Incremental Transformer to encode multi-turn utterances along with knowledge in related documents. Recently, Dinan et al. (2019) propose both a dataset of Wizard of Wikipedia and a model to leverage the two-step procedure of selecting knowledge from the pool and generating a response based on chosen knowledge and given context.

One of the most related models to ours may be Lian et al. (2019), who also focus on the knowledge selection issue in the two-stage knowledge-grounded dialogue. However, our work is novel in that we model it as a sequential decision process with latent variables and introduce the knowledge loss. Thanks to these updates, our model achieves significantly better performance as shown in the experiments.

**Sequential Latent Variable Models**. There have been many studies about sequential latent variable models. Chung et al. (2015) propose one of the earliest latent models for sequential data, named VRNN. Later, this architecture is extended to SRNN (Fraccaro et al., 2016) and Z-Forcing (Goyal et al., 2017). There have been some notable applications of sequential latent models, including doc-

| Seen Test (Topic: Italian Cuisine) | Unseen Test (Topic: Hunting) |
|---|---|
| ⋮ | ⋮ |
| A: I love chicken parmigiana as well, but I think my ultimate favorite is beef lasagna… Extra cheese please!
W: Chicken with sauce and mozzarella…. Be still my heart!
A: Truthfully, anything with cheese is the best | W: That is true but we always have to watch out for excessive hunting. It has caused some species to be endangered.
A: Yes I agree. I don't believe in the useless hunting that poachers do. Its so cruel. |
| **(Ours)** i love pizza too ! it's a traditional italian dish consisting of yeasted flatbread typically topped with tomato sauce and cheese
**(TMN)** i love cheese !
**(E2E BERT+KL)** i like mine topped with vegetables , meats , and condiments .
**(Human)** especially cheddar cheese ! it's the second most popular cheese in the use ! | **(Ours)** i agree , poaching has been defined as the illegal hunting or capturing of wild animals .
**(TMN)** i thinks so , i'm not sure if you're talking about poaching , but i know that poodles are the second most intelligent breed behind the poodle .
**(E2E BERT+KL)** i agree . i think it's a great way to catch fish .
**(Human)** agreed , i remember reading one time that unless you plan to kill the animals its not considered hunting . |

Figure 3: Examples of generated responses by our model and baselines on Wizard of Wikipedia. TMN stands for E2E Transformer MemNet, and A and W for apprentice and wizard. Examples with selected knowledge sentences can be found at Appendix E.

ument summarization (Li et al., 2017), image captioning (Aneja et al., 2019) and text generation (Shao et al., 2019). Another related class of sequential latent models may be *latent attention models* (Deng et al., 2018; Wang et al., 2018; Yang et al., 2017), which exploit the latent variables to model the attention mapping between input and output sequences. Although our method is partly influenced by such recent models, it is novel to propose a sequential latent model for the knowledge-grounded chit-chat problem.

## 6 CONCLUSION

This work investigated the issue of knowledge selection in multi-turn knowledge-grounded dialogue, and proposed a sequential latent variable model, for the first time, named *sequential knowledge transformer* (SKT). Our method achieved the new state-of-the-art performance on the Wizard of Wikipedia benchmark (Dinan et al., 2019) and a knowledge-annotated version of Holl-E dataset (Moghe et al., 2018). There are several promising future directions beyond this work. First, we can explore other inference models such as sequential Monte Carlo methods using *filtering variational objectives* (Maddison et al., 2017a). Second, we can study the interpretability of knowledge selection such as measuring the uncertainty of attention (Heo et al., 2018).

### ACKNOWLEDGMENTS

We thank Hyunwoo Kim, Chris Dongjoo Kim, Soochan Lee, Junsoo Ha and the anonymous reviewers for their helpful comments. This work was supported by SK T-Brain corporation and Institute of Information & communications Technology Planning & Evaluation (IITP) grant funded by the Korea government (MSIT) (No.2019-0-01082, SW StarLab). Gunhee Kim is the corresponding author.

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

## A    DERIVATION OF CONDITIONAL PROBABILITY

In Section 3, we re-write the conditional probability of wizard's response $\mathbf{y}^t$ given dialogue context $\mathbf{x}^{\leq t}$ and $\mathbf{y}^{<t}$ from Eq. (2) to Eq. (4). We can simply derive it as follows:

$$p(\mathbf{y}|\mathbf{x}) \tag{13}$$

$$= \prod_t \sum_{\mathbf{k}^t} p_\theta(\mathbf{y}^t|\mathbf{x}^{\leq t}, \mathbf{y}^{<t}, \mathbf{k}^{\leq t})\pi_\theta(\mathbf{k}^t) \quad \text{(by Eq. (2))} \tag{14}$$

$$= \prod_{i=1}^{t-1} \sum_{\mathbf{k}^i} p_\theta(\mathbf{y}^i|\mathbf{x}^{\leq i}, \mathbf{y}^{<i})p_\theta(\mathbf{k}^i)\Big(\sum_{\mathbf{k}^t} p_\theta(\mathbf{y}^t|\mathbf{x}^{\leq t}, \mathbf{y}^{<t}, \mathbf{k}^{\leq t})\pi_\theta(\mathbf{k}^t)\Big) \quad \text{(by Bayes' rule)} \tag{15}$$

$$\approx \prod_{i=1}^{t-1} \sum_{\mathbf{k}^i} p_\theta(\mathbf{y}^i|\mathbf{x}^{\leq i}, \mathbf{y}^{<i})q_\phi(\mathbf{k}^i)\Big(\sum_{\mathbf{k}^t} p_\theta(\mathbf{y}^t|\mathbf{x}^{\leq t}, \mathbf{y}^{<t}, \mathbf{k}^{\leq t})\pi_\theta(\mathbf{k}^t)\Big) \tag{16}$$

$$= \prod_{i=1}^{t-1} \sum_{\mathbf{k}^i} q_\phi(\mathbf{k}^i)\Big(\sum_{\mathbf{k}^t} p_\theta(\mathbf{y}^t|\mathbf{x}^{\leq t}, \mathbf{y}^{<t}, \mathbf{k}^t)\pi_\theta(\mathbf{k}^t)\Big) \quad (\mathbf{x}^{\leq t} \text{ and } \mathbf{y}^{<t} \text{ are given}) \tag{17}$$

$$\approx p(\mathbf{y}^t|\mathbf{x}^{\leq t}, \mathbf{y}^{<t}), \tag{18}$$

where $q_\phi(\mathbf{k}^i)$ is an approximated posterior distribution and $p_\theta(\mathbf{k}^i)$ is a true posterior distribution.

## B    TRAINING DETAILS

All the parameters except pretrained parts are initialized with Xavier method (Glorot & Bengio, 2010). We use Adam optimizer (Kingma & Ba, 2015) with $\beta_1 = 0.9, \beta_2 = 0.999, \epsilon = 1e-07$. For the models without BERT, we set the learning rate to 0.001 and initialize the embedding matrix with `fastText` (Bojanowski et al., 2016) trained on the Common Crawl corpus. For the models with BERT, we set the learning rate to 0.00002 and initialize encoder weights with `BERT-Base, Uncased` pretrained weights. We apply label smoothing (Pereyra et al., 2017; Edunov et al., 2017; Vaswani et al., 2017) for both knowledge selection and response generation, and set 0.1 and 0.05 for each. We set the temperature of Gumbel-Softmax to $\tau = 0.1$ and the hyperparameter for the knowledge loss to $\lambda = 1.0$. For efficiency, we batch the dialogues rather than individual turns. We train our model up to 5 epochs on two NVIDIA TITAN Xp GPU.

## C    KNOWLEDGE SELECTION ACCURACY OVER TURNS

Table 6 compares the knowledge selection accuracy of different methods for each turn on the Wizard of Wikipedia. Thanks to the sequential latent variable, our model consistently outperforms other methods for all turns in knowledge selection accuracy. Notably, in all models, the accuracy significantly drops after the first turn, which is often easily predictable as a topic definition sentence. It shows the diversity nature in knowledge selection, as discussed in Section 2.

## D    QUANTITATIVE RESULTS ON SEMI-SUPERVISED SETTING

Table 7 shows the results of our model with partial knowledge labels on the Wizard of Wikipedia. We attain better performance with more labeled knowledge data for training as expected. Furthermore,

Table 6: Knowledge selection accuracy for each turn on the Wizard of Wikipedia (Dinan et al., 2019). The method with [*] uses no knowledge loss. TMN stands for E2E Transformer MemNet.

| Method | Test Seen | | | | | Test Unseen | | | | |
|---|---|---|---|---|---|---|---|---|---|---|
| | 1st | 2nd | 3rd | 4th | 5th | 1st | 2nd | 3rd | 4th | 5th |
| PostKS* (Lian et al., 2019) | 3.6 | 3.6 | 4.1 | 7.0 | 9.5 | 3.4 | 3.0 | 4.7 | 4.1 | 9.9 |
| PostKS + Knowledge Loss | 55.4 | 19.3 | 10.7 | 8.7 | 7.0 | 26.0 | 3.8 | 4.0 | 3.9 | 3.8 |
| TMN (Dinan et al., 2019) | 55.8 | 19.5 | 10.4 | 7.6 | 6.2 | 25.9 | 7.0 | 4.1 | 4.2 | 6.1 |
| E2E BERT + PostKS | 56.5 | 20.6 | 13.7 | 10.4 | 9.2 | 36.0 | 8.1 | 6.1 | 6.8 | 5.7 |
| Ours | 59.1 | 20.6 | 15.8 | 12.8 | 9.1 | 52.9 | 8.8 | 8.4 | 6.4 | 10.7 |

Table 7: Performance of our model with partial knowledge labels on Wizard of Wikipedia (Dinan et al., 2019).

| Method | Test Seen | | | | Test Unseen | | | |
|---|---|---|---|---|---|---|---|---|
| | PPL | R-1 | R-2 | Acc | PPL | R-1 | R-2 | Acc |
| E2E Transformer MemNet[†] (Dinan et al., 2019) | 63.5 | 16.9 | - | 22.5 | 97.3 | 14.4 | - | 12.2 |
| E2E Transformer MemNet (BERT vocab)[‡] | 53.2 | 17.7 | 4.8 | 23.2 | 137.8 | 13.6 | 1.9 | 10.5 |
| Ours | 52.0 | 19.3 | 6.8 | 26.8 | 81.4 | 16.1 | 4.2 | 18.3 |
| 1/2 knowledge labeled | 49.0 | 19.2 | 6.6 | 25.1 | 77.8 | 16.1 | 4.1 | 16.7 |
| 1/4 knowledge labeled | 45.7 | 18.7 | 6.1 | 22.4 | 78.0 | 15.8 | 3.6 | 13.8 |
| 1/8 knowledge labeled | 45.3 | 18.6 | 6.0 | 21.0 | 79.9 | 15.7 | 3.6 | 12.3 |
| no knowledge loss | 54.7 | 17.1 | 4.6 | 0.3 | 88.2 | 15.5 | 3.4 | 0.1 |

our model achieves competitive performance with less label. For instance, our model using only 1/4 labeled training data is comparable to E2E Transformer MemNet and even better in Test Unseen. As a result, our sequential latent knowledge selection model can be utilized in a semi-supervised method without severe drop in the performance.

# E    EXAMPLES WITH SELECTED KNOWLEDGE

Figure 4 and 5 show selected examples of knowledge selection and response generation. In each set, given dialogue context, we compare selected knowledge and generated utterances by our method and baselines with human ground truths.

| **Seen Test (Topic: Italian Cuisine)** | |
| --- | --- |
| ⋮ | |
| A: I love chicken parmigiana as well, but I think my ultimate favorite is beef lasagna… Extra cheese please! | |
| W: Chicken with sauce and mozzarella…. Be still my heart! | |
| A: Truthfully, anything with cheese is the best | |

| | **Task 1: Knowledge Selection** | **Task 2: Response Generation** |
| --- | --- | --- |
| **Ours** | Pizza is a traditional Italian dish consisting of a yeasted flatbread typically topped with tomato sauce and cheese and baked in an oven. (from "Pizza" Wikipedia page) | i love pizza too ! it ' s a traditional italian dish consisting of yeasted flatbread typically topped with tomato sauce and cheese |
| **TMN** | no_passages_used | i love cheese ! |
| **E2E BERT +KL** | It can also be topped with additional vegetables, meats, and condiments, and can be made without cheese. (from "Pizza" Wikipedia page) | i like mine topped with vegetables , meats , and condiments . |
| **Human** | It is the second-most popular cheese in the US (behind mozzarella), with an average annual consumption of 10 lb (4.5 kg) per capita. (from "Cheddar cheese" Wikipedia page) | especially cheddar cheese ! it ' s the second most popular cheese in the use ! |

Figure 4: Examples of selected knowledge and generated responses by our model and baselines on the WoW Seen Test set.

| **Unseen Test (Topic: Hunting)** | |
| --- | --- |
| ⋮ | |
| A: Yes, that is the best way to do it. Apparently in some areas the government will actually pay money to hunt for … | |
| W: That is true but we always have to watch out for excessive hunting. It has cause some species to be endangered. | |
| A: Yes I agree. I don't believe in the useless hunting that poachers do. It's so cruel. | |

| | **Task 1: Knowledge Selection** | **Task 2: Response Generation** |
| --- | --- | --- |
| **Ours** | Poaching has traditionally been defined as the illegal hunting or capturing of wild animals, usually associated with land use rights. (from "Poaching" Wikipedia page) | i agree , poaching has been defined as the illegal hunting or capturing of wild animals . |
| **TMN** | no_passages_used | i thinks so , i ' m not sure if you ' re talking about poaching , but i know that poodles are the second most intelligent breed behind the poodle . |
| **E2E BERT +KL** | Hunting can also be a means of pest control. (from "Poaching" Wikipedia page) | i agree . i think it ' s a great way to catch fish . |
| **Human** | It is also not considered hunting to pursue animals without intent to kill them, as in wildlife photography, birdwatching, or scientific research (from "Hunting" Wikipedia page) | agreed , i remember reading one time that unless you plan to kill the animals its not considered hunting . |

Figure 5: Examples of selected knowledge and generated responses by our model and baselines on the WoW Unseen Test set.