# OpenReview forum: "Sequential Latent Knowledge Selection for Knowledge-Grounded Dialogue"
_ICLR.cc/2020/Conference — Accept (Spotlight)_

### Official Review · AnonReviewer2 · 2019-10-21
**Official Blind Review #2**

**Rating:** 6

**Review:**

This paper presents a sequential latent variable model for knowledge selection in dialogue generation. More specifically, the authors extended the posterior attention model (Shankar and Sarawagi, 2019) to the latent knowledge selection problem. The proposed model achieved higher performances than previous state-of-the-art knowledge-grounded dialogue models on Wizard of Wikipedia and Holl-E datasets.

This work presents a reasonable ideas with new state-of-the-art results in both quantitative and qualitative evaluations.
And overall the paper reads well.

But I think it could be further improved with the following points:
- Could you describe the updates from the previous sequential latent variable models more clearly? It would help to further highlight the contribution of this work. Now it might not be very clear enough for those who are not familiar with the previous work.
- In the introduction, the authors claim the following three advantages of the proposed method: reduced scope of knowledge candidates, better utilization of response information, and weakly-supervised inference with no labels.
But I'm not very convinced whether the experimental results indicate the aspects clearly enough. More detailed analysis should be added to support the contributions.
- The current experiments mainly focus on end-to-end dialogue generation performances. But it would be also interesting to see more detailed aspects of knowledge-selection itself in both quantitative and qualitative manners. I guess this analysis can be done based on the sampled or selected knowledge from the attention distribution.
- Could you possibly add some ablation studies to show the effectiveness of each component? Especially, I'm curious about the results of the proposed model without knowledge loss.

**Experience Assessment:**

I have published one or two papers in this area.

**Review Assessment: Checking Correctness Of Derivations And Theory:**

I assessed the sensibility of the derivations and theory.

**Review Assessment: Checking Correctness Of Experiments:**

I carefully checked the experiments.

**Review Assessment: Thoroughness In Paper Reading:**

I read the paper at least twice and used my best judgement in assessing the paper.

---

> ### Author Response · Authors · 2019-11-15
> **Response to Reviewer 2**
>
> We thank Reviewer 2 for positive and constructive reviews. Below, we respond to each comment in detail. Please see the blue fonts in the newly uploaded draft to check how our paper is updated.
>
>
> 1. Could you describe the updates from the previous sequential latent variable models more clearly?
>
> Our main contribution is to model the knowledge selection in dialogue as sequential latent variable models for the first time, and validate that it leads to the new state-of-the-art performances on two benchmark datasets. The use of sequential latent models can correctly deal with diversity nature in knowledge selection in a semi-supervised way and improves the interpretability of the flow of selected knowledge over other models. Methodologically, our model is similar to [1], although it uses the latent variable to represent the underlying attention in seq2seq models for machine translation (unlike ours for knowledge-grounded chit-chat problem).
>
> [1] S. Shankar and S. Sarawagi. Posterior Attention Models for Sequence to Sequence Learning. ICLR 2019.
>
>
> 2. Ablation studies for the three advantages of the proposed method: (i) weakly-supervised inference with no labels, (ii) reduce the scope of knowledge candidates, and (iii) better utilization of response information.
>
> We here answer the reviewer’s second and fourth questions together.
>
> We add the experiments of our model with partial knowledge labels (including an experiment without knowledge loss) on the Wizard of Wikipedia in Table 6 in Appendix D. Results show that the better performance is attained with more labeled knowledge data for training as expected. Furthermore, our model achieves competitive performance with less label. For instance, our model using only 1/4 labeled training data is comparable to E2E Transformer MemNet and is even better in Test Unseen. As a result, our sequential latent knowledge selection model can be utilized in a semi-supervised method without severe drop in its performance.
>
> Due to many new experiments during the limited time of rebuttal, we cannot finish an ablation study for the reduced scope and utilization of response information, which will be presented in the final draft.
>
>
> 3. It would be also interesting to see more detailed aspects of knowledge-selection itself in both quantitative and qualitative manners.
>
> We add more quantitative and qualitative results of knowledge selection in Appendix C and G. In Appendix C, we measure the knowledge selection accuracy over turns. Our model consistently outperforms other models for all turns in knowledge selection accuracy. Notably, in all models, the accuracy significantly drops after the first turn (which is often easily predictable topic definition sentence), which shows the diversity nature in knowledge selection. In Appendix G, we show selected examples of utterance prediction along with selected knowledge. We will update more qualitative results (e.g. attention distribution) to our final version.

---

### Official Review · AnonReviewer1 · 2019-10-23
**Official Blind Review #1**

**Rating:** 8

**Review:**

Post author response edit: The authors did a good job of addressing many of the concerns of reviewers. I believe with these new results (esp to reviewer 4), they will have a stronger version for the camera ready. I'm bumping up my recommendation for this reason.



The authors propose a novel architecture for selecting knowledge in knowledge-grounded multi-turn dialogue. Their knowledge selection module uses a sequential latent variable scheme, and is claimed to be able to both handle diversity of knowledge selection in conversation as well as leverage the information from the response. The proposed model yields state of the art on two relevant benchmark datasets in terms of perplexity and F1, and scores higher in human evaluations as well.

The paper is relatively well-written, and the authors offer extensive insight into their approach, providing relevant equations and diagrams where necessary. The approach is well-motivated, and the experiments indicate that the model indeed helps on all evaluation fronts. A variety of baselines are considered and are shown to be inferior, in nearly every metric. I did not spend a lot of effort to try to understand their factorization, but the intuition makes sense, and their use of gumbel softmax provides a clear avenue to fix some of the hard-backprop issues apparent in the original Dinan et al. paper. I also appreciate the addition of the knowledge loss to the PostKS baseline: it’s a good effort to make the baseline as good as possible.

A few things bother me with the paper. The primary one is it concerns me a bit that the BERT pretraining does not improve significantly over the E2E transformer memnet (with just bert vocabulary). Unless I’m missing something, that model contained NO pretraining, so I would expect massive improvements. A sanity check there would be checking ppl with gold knowledge: if that doesn’t significantly improve, then I suspect the authors have something really weird about the pretraining or fine tuning. However, It also appears to me that replacing the GRU with a transformer in PostKS might be unfair: Transformers are way more data hungry than RNNs, and so both variants should be tried (though I would be okay with the loser being relegated to a footnote or appendix).

The human evaluations are not as convincing as the authors propose them to be, especially the difference in the “Test Seen” case. It is unclear to me why the authors believe that their “model’s merit would be more salient” in a multi-turn setting, and I think such an experiment would be good to show - or, at the very least, an indication that such an experiment was tried but results were not considered due to reasons X,Y, Z, etc. Overall, the rough improvement that is being provided in the first-stage of the two stage setting seems rather minor (23% -> 26% accuracy; 2.21 -> 2.35 human eval), and that the task remains extremely difficult

Questions
* I know the Dinan et al. models, at human evaluation time, hardcoded to not pick the same knowledge twice. Do you have a similar restriction? If not, maybe you can at least say that you manage to get rid of the need for that!
* As mentioned earlier, I would be curious to see multi-turn human evaluations. I understand this is expensive and a large ask.
* How are the examples in figure 3 chosen? Are they generally indicative of what is seen throughout the human evaluation?
* It would be useful to see a qualitative example of the model’s knowledge selection process when comparing to other models, rather than just the utterance generation (which is not the novel contribution of the paper).

Nits
* Small grammatical errors dealing with subject-verb agreement (plurals mostly).
* Using “-” instead of n/a in tables would make it mildly easier to see digest and see where metrics don’t make sense.


**Experience Assessment:**

I have published in this field for several years.

**Review Assessment: Checking Correctness Of Derivations And Theory:**

I did not assess the derivations or theory.

**Review Assessment: Checking Correctness Of Experiments:**

I assessed the sensibility of the experiments.

**Review Assessment: Thoroughness In Paper Reading:**

I read the paper at least twice and used my best judgement in assessing the paper.

---

> ### Author Response · Authors · 2019-11-15
> **Response to Reviewer 1**
>
> We thank Reviewer 1 for positive and constructive reviews. Below, we respond to each comment in detail. Please see blue fonts in the newly uploaded draft to check how our paper is updated.
>
>
> 1. Sanity check for BERT implementation & Reason for the low performance of BERT
>
> Following your suggestion, we measure our model’s performance with gold knowledge. The table below shows that providing gold knowledge significantly improves our model’s performance. It can be a good sanity check for our implementation.
>
> 				Test Seen				Test Unseen
> Method			PPL		R-1		R-2		PPL		R-1		R-2
> Ours			52.0		19.3		6.8		81.4		16.1		4.2
> Ours (w/ gold)	23.1		34.2		18.4		27.8		32.6		16.6
>
> The reason for the low performance of BERT may be the diversity nature of knowledge selection in knowledge-grounded dialogue. As discussed in Section 2, there can be one-to-many relations between the dialogue context and the knowledge to be selected. One can choose any diverse knowledge to carry on the conversation. Table 1 confirms this conjecture. In the Wizard of Wikipedia dataset, knowledge selection is extremely challenging even for human (17.1) and BERT is marginally better than Transformer (23.4 of BERT vs 22.5 of Transformer). On the other hand, once we change the task to have one-to-one relations by providing a GT response, BERT significantly boosts performance over Transformer (78.2 in BERT vs 70.4 in Transformer).
>
>
> 2. Quantitative results of PostKS with GRU
>
> We add quantitative results of PostKS+Transformer and PostKS+GRU on the Wizard of Wikipedia and Holl-E in Table 7 in Appendix E. Results show that PostKS+GRU consistently outperforms PostKS+Transformer without the knowledge loss term. The lower performance of PostKS+Transformer may be due to the data starvation problem as the review anticipated. However, PostKS+Transformer performs better than PostKS+GRU with the knowledge loss. It seems that the knowledge loss term reduces the overfitting and thus increases the data efficiency.
>
>
> 3. Multi-turn human evaluation results
>
> We add human evaluation results in a multi-turn setting using the evaluation toolkit from Wizard of Wikipedia. Following their setting, humans are paired with one of the models and chat about a specific topic (given a choice of 2-3 topics) for 3-5 dialogue turns. After conversation, they rate their dialogue partner on a scale of 1-5, with the rating indicating how much they “liked” the conversation. We collect the votes for 110 randomly sampled conversations from 10 different turkers.
>
> Models						Test Seen	Test Unseen
> E2E Transformer MemNet		2.36 (1.38)	2.10 (0.96)
> Ours						2.39 (0.99)	2.38 (1.01)
>
> As shown in the table (avg and stddev), human annotators prefer our results to those of baselines with a larger gap in Test Unseen.
>
>
> 4. Overall, the rough improvement that is being provided in the first-stage of the two stage setting seems rather minor (23% -> 26% accuracy; 2.21 -> 2.35 human eval), and that the task remains extremely difficult.
>
> Considering the difficulty of the task, our improvement in knowledge selection (23.2% -> 26.8%) is not minor. Dinan et al. (2019) recorded 25.5% accuracy in knowledge selection on WoW with additional 700 million Reddit conversations [1] and knowledge selection data of [2], while ours achieves better performance even without them. Agreeing that the task remains challenging, we strongly believe that our work brings important contributions for knowledge-ground conversation: (i) focusing the diversity issue of knowledge selection for the first time, (ii) correctly modeling it as a sequential latent model and (iii) achieving new state-of-the-art performance with nontrivial margins.
>
> [1] P. Mazare, S. Humeau, M. Raison, and A. Bordes. Training Millions of Personalized Dialogue Agents. EMNLP, 2018.
> [2] P. Rajpurkar, J. Zhang, K. Lopyrev, and P. Liang. SQuAD: 100,000+ Questions for Machine Comprehension of Text. EMNLP, 2016.
>
>
> 5. I know the Dinan et al. models, at human evaluation time, hardcoded to not pick the same knowledge twice. Do you have a similar restriction? If not, maybe you can at least say that you manage to get rid of the need for that!
>
> Thank you for your suggestion. We did not use their hardcoding. We will add that statement to our final version.
>
>
> 6. How are the examples in figure 3 chosen? Are they generally indicative of what is seen throughout the human evaluation?
>
> We manually select one example for Figure 3. For human evaluation, we randomly sample test examples without knowing which examples are chosen at all. We will add more examples of knowledge selection and utterance prediction in Appendix G.
>
>
> 7. Qualitative examples with selected knowledge
>
> Thank you for your suggestion. We add qualitative examples of selected knowledge in Appendix G.
>
>
> 8. Grammatical errors
>
> We update our paper per your suggestion.

---

### Official Review · AnonReviewer4 · 2019-11-01
**Official Blind Review #4**

**Rating:** 8

**Review:**

The paper looks at the problem of knowledge selection for open-domain dialogue. The motivation is that selecting relevant knowledge is critical for downstream response generation.
The paper highlights the one-to-many relations when selecting knowledge which makes the problem even more challenging. It tries to address this by taking into account the history of knowledge selected at previous turns.
The paper proposes a Sequential Latent Model which represents the knowledge history as some latent representation. From this methodology they select a piece of knowledge at the current turn and use it to decode an utterance. The model is trained in a joint fashion to learn which knowledge to select and on generating the response. As the two are strongly correlated. Additionally there is an auxiliary loss to help correctly identify if the knowledge was correctly selected. Additionally a copy mechanism is introduced to try to copy words from the knowledge during decoding.
The experiments are run on the Wizard of Wikipedia dataset where there are annotations for which knowledge sentence is selected and on Holl-E, where they transform the dataset to have a single sentence tied to a response.
For automatic metrics there is significant improvement over baselines for correctly selecting a piece of knowledge and generating a response. Additionally there is human evaluation that also shows significant improvement.  Their model also seems to generalize well to domains that were not seen during training time over baselines models.

The contribution of the paper is the novel approach to selecting knowledge for open-domain dialogue. This work is significant in that by improving knowledge selection we see a subsequent improvement in response generation quality which is the overall downstream task within this problem space.
I believe this paper should be accepted because of the significant and novel approach of modeling previous knowledge sentences selected. The linking of this knowledge selection model to topic tracking as stated in the paper is of clear importance, as ensuring topical depth and topical transition are two key aspects for open-domain dialog.

Feedback on the paper
In Figure 3, please provide the knowledge sentence that was selected.
Please provide the inter-annotator agreement for human evaluation.
I think it would be interesting to see what is the copy mechanism actually adding in terms of integration of knowledge vs the WoW MemNet approach. Are those two truely comparable because one does not have copy?
For Related Work, also cite Topical-Chat: Towards Knowledge-Grounded Open-Domain Conversations

Small grammatical errors
"Recently, Dinan et al. (2019) propose to tackle" -> "Recently, Dinan et al. (2019) proposed to tackle"
"which subsequently improves the knowledge-grounded chit-chat." -> "which subsequently improves knowledge-grounded chit-chat."


Some questions for the authors in terms of future direction
How is the performance of the model impacted with longer dialog context vs shorter?

The Holl-E dataset was transformed from spans of knowledge to a single knowledge sentence. It would be interesting to see what happens when the knowledge selected is over multiple sentences.

The knowledge pool currently consists of 67.57 sentences on average. How will this method scale as the amount of knowledge sentences grows?






**Experience Assessment:**

I have published one or two papers in this area.

**Review Assessment: Checking Correctness Of Derivations And Theory:**

I assessed the sensibility of the derivations and theory.

**Review Assessment: Checking Correctness Of Experiments:**

I carefully checked the experiments.

**Review Assessment: Thoroughness In Paper Reading:**

I read the paper at least twice and used my best judgement in assessing the paper.

---

> ### Author Response · Authors · 2019-11-15
> **Response to Reviewer 4**
>
> We thank Reviewer 4 for positive and constructive reviews. Below, we respond to each comment in detail. Please see the blue fonts in the newly uploaded draft to check how our paper is updated.
>
>
> 1. In Figure 3, please provide the knowledge sentence that was selected.
>
> Thanks for the suggestion. We add some examples of selected knowledge and predicted utterances in Appendix G.
>
>
> 2. Please provide the inter-annotator agreement for human evaluation.
>
> We measured the agreement among the annotators using Fleiss’ kappa [1]. All kappa values exceeded or were close to 0.2, indicating the slight agreement among annotators. There were some diversity among annotators’ responses, because of utilizing the 4 point scale in order to avoid having a “catch-all” category (i.e. no middle response scale) in the answer choice [2]. To mitigate such annotator bias and inter-annotator variability, we adjusted human evaluation results via Bayesian calibration [3]. Table 4 shows raw and calibrated results of human evaluation, which consistently validates that annotators prefer our results to those of the baselines.
>
> 					Test Seen						Test Unseen
> Method		Engagingness	Knowledgeability	Engagingness	Knowledgeability
> PostKS		0.12				0.17				0.12				0.09
> TMN		0.22				0.19				0.16				0.17
> Ours		0.20				0.20				0.21				0.17
> Human		0.22				0.22				0.23				0.31
>
> [1] J. L Fleiss. Measuring Nominal Scale Agreement among Many Raters. Psychol. Bull. 1971.
> [2] D. K. Dalal, N. T. Carter, and C. J. Lake. Middle Response Scale Options are Inappropriate for Ideal Point Scales. J. Bus. Psychol. 2014.
> [3] I. Kulikov, A. H. Miller, K. Cho, and J. Weston. Importance of Search and Evaluation Strategies in Neural Dialogue Modeling. INLG 2019.
>
>
> 3. I think it would be interesting to see what is the copy mechanism actually adding in terms of integration of knowledge vs the WoW MemNet approach.
>
> In newly updated draft, we add quantitative results of “E2E Transformer MemNet + BERT + PostKS + Copy” to Table 2 and 3. To make the results more reliable, we run the model three times with different random seeds and report its mean. We also update our model’s results in the same manner. The “E2E Transformer MemNet + BERT + PostKS + Copy” performs the best among baselines, but is not as good as ours, which confirms that sequential latent modeling is critical for improving the accuracy of knowledge selection and subsequently utterance generation. Adding the copy mechanism to the baseline substantially improves the accuracy of utterance generation, but barely improves the knowledge selection accuracy, which also justifies the effectiveness of the sequential latent variable. Additionally, the performance gaps between ours and baselines are larger in Test Unseen. It can be understood that the sequential latent variable can generalize better.
>
>
> 4. For Related Work, also cite Topical-Chat: Towards Knowledge-Grounded Open-Domain Conversations.
>
> Thank you. We update our paper as your suggestion.
>
>
> 5. How is the performance of the model impacted with longer dialog context vs shorter?
>
> Table 5 in Appendix C compares the knowledge selection accuracy of different methods for each turn on the Wizard of Wikipedia. Thanks to the sequential latent variable, our model consistently outperforms other models for all turns in knowledge selection accuracy. Notably, in all models, the accuracy significantly drops after the first turn (which is often easily predictable topic definition sentence), which shows the diversity nature in knowledge selection, as discussed in Section 2.
>
>
> 6. The Holl-E dataset was transformed from spans of knowledge to a single knowledge sentence. It would be interesting to see what happens when the knowledge selected is over multiple sentences.
>
> We select the sentence that includes the span as the ground-truth (GT) knowledge sentence. If the span is given over multiple sentences, we select the minimum number of consecutive sentences containing the span and use them as GT. If all of the candidate sentences have zero F1 scores to the span and the response, we tag ‘no_passages_used’ as the GT, which amounts to 5% of GT labels. All of the details are updated in Section 4.1.
>
>
> 7. The knowledge pool currently consists of 67.57 sentences on average. How will this method scale as the amount of knowledge sentences grows?
>
> Due to the use of BERT (or Transformer) as the sentence encoder, our memory complexity is O(nm^2) where n is the number of candidate sentences in knowledge pool and m is the length of the longest sentence. For example, when training with 1 dialogue batch on NVIDIA TITAN RTX GPU, our model scales up to n=95 and m = 32. But at test time, it is highly scalable up to n>=500 because of no need for backpropagation.
>
>
> 8. Grammatical errors
>
> Thank you. We update our paper as your suggestion.

---

### Decision · Program_Chairs · 2019-12-19

**Decision:**

Accept (Spotlight)

**Comment:**

This paper proposes a sequential latent variable model for the knowledge selection task for knowledge grounded dialogues. Experimental results demonstrate improvements over the previous SOTA in the WoW, knowledge grounded dialogue dataset, through both automated and human evaluation. All reviewers scored the paper highly, but they also made several suggestions for improving the presentation. Authors responded positively to all these suggestions and provided updated results and other stats. The paper will be a good contribution to ICLR.